# Beyond “Doing Better”: Ordinal Rating Scales to Monitor Behavioural Indicators of Well-Being in Cats

**DOI:** 10.3390/ani12212897

**Published:** 2022-10-22

**Authors:** Jacklyn J. Ellis

**Affiliations:** Toronto Humane Society, Toronto, ON M5J 4C5, Canada; jellis@torontohumanesociety.com

**Keywords:** cats, welfare, well-being, quality of life, behaviour assessment, ordinal rating scale, interobserver, shelter

## Abstract

**Simple Summary:**

Animal shelters need to do their best to ensure that the cats in their care have a good quality of life. There are many strategies they can use to meet this goal. However, it is important to evaluate if these strategies are successful to ensure continual improvement of care. This evaluation can be challenging, as tools for this purpose are limited. This paper presents a method of evaluating four different expressions of cat welfare on a 0–5 scale and shows that results are very similar between raters (after a training program), ensuring that the data are reliable. These rating scales may prove useful for other shelters or any environment where objective evaluation of cat welfare is required.

**Abstract:**

Safeguarding the well-being of cats is essential to the mission of any responsible animal shelter. Environmental enrichment and behaviour modification are often key to this goal. Measuring response to these interventions is essential to ensure strategies are successful. There are often many staff and volunteers involved in these efforts, and a lack of standardised language can make monitoring progress difficult. Ordinal rating scales of key behaviours can be a useful way to summarise observations and ensure that common language is used. However, it is crucial that these scales have good interobserver agreement and reliability, so operational definitions and training systems are important. This paper presents a method for evaluating four different expressions of cat welfare on a 0–5 scale: modified Fear, Anxiety, and Stress score; Response to Petting score; Participation in Play score; and Food Intake Summary score. All scales showed almost perfect average interobserver agreement (linear weighted κ) and excellent average interobserver reliability (interclass correlation coefficient). These scales may prove useful to other shelters, or any other environment where evaluating response to interventions is important to the welfare of cats, such as research facilities or home environments. The exceptional interobserver agreement and reliability of this study compared with some others highlights the importance of standardised training programs.

## 1. Introduction

Animal shelters have a responsibility to safeguard the psychological well-being of the cats in their care. Recommended strategies for this often include minimizing exposure to environmental stressors [1], enriching their enclosure [2], and behaviour modification plans [3]. To optimize these efforts, it is necessary to assess the impact they are having on the animal, for example by monitoring changes in behaviour across time and in response to specific interventions [4].

Often there are multiple staff and volunteers involved in implementing these strategies, and unfortunately the language used to communicate observations can vary in quality and suffer from a lack of standardisation. Reports can include wording such as “they are doing better” or “it was a good day for them”, and these phrases lack the specificity and detail required for a shared understanding of what aspects of their behaviour are improving and how behaviour is changing over time. These pieces of information are required to properly evaluate if the interventions correlate with positive changes to behavioral indicators of well-being, or how the impacts of one intervention compare with the impacts of another in similar cases.

To resolve these concerns, it would be ideal to standardise the language used and conditions under which observations are recorded. Many of the methods used in academic research are impractical for regular use in most shelters, due to the time and technical skills/equipment required (e.g., continuous behaviour observation via 24 h video recording analysed using Noldus^®^ Observer). Ordinal rating systems are appealing for this purpose, as they can represent various levels of intensity, can be measured live, can efficiently summarise a lot of data, and can easily reveal patterns of improvement or deterioration [5]. Within the field of assessing well-being in cats, ordinal data are already being used. The Cat-Stress-Score (CSS) [6] is ubiquitous in the literature and sets a good precedent for using this type of data across time and individual raters to communicate objective information about a cat’s behaviour and emotional state.

When multiple observers use an ordinal rating system to communicate an animal’s behaviour, it is essential that all raters have a common understanding of each level of the scale and that the ratings of each observer agree with an accepted reference standard rating. For example, the CSS has been reported to have interobserver agreement as high as 0.9 [6] or as low as 0.29 [7]. One reason for the discrepancy in agreement reported for this scale can be the level of training provided to raters. Few studies report how much training is provided. Kessler and Turner [6] reported interobserver agreement of 0.9 when two trained observers were used, but 0.75 among staff with less training. Unfortunately, there is no published training guide available for the CSS.

In shelters, sometimes cats continue to exhibit very stressed body language, but begin to respond to petting, show tentative signs of play, or increase their food intake. In this case, simply measuring behavioural and postural signs of stress would miss these other signs of progress. A small number of carefully selected ordinal rating scores collected at standardised times of the day (e.g., during cleaning or behaviour modification sessions) would produce a more holistic picture of the cat’s psychological well-being

Accordingly, four ordinal scales were adapted or designed by the author for the purposes of assessing different behavioural aspects of a cat’s well-being. First, the Fear, Anxiety, and Stress (FAS) score, from the Fear-Free Pets program [8], was adapted to assess stress from the cat’s body language. The FAS was elected over the CSS because a similar (unpublished) adaptation was made to the dog FAS score for use at our facility (while no dog system comparable to the CSS exists). Second, a Response to Petting (RTP) score was designed to convey the cat’s level of comfort with human contact. Third, a Participation in Play (PIP) score was designed to convey the enthusiasm with which the cat engages in play. Finally, a Food Intake Summary (FIS) score was developed to summarise all available information about the amount of food the cat consumed in the 24 h prior to assessment. The decision to measure these four parameters developed organically during the monitoring of different cases in shelter, as progress in different cats was observed that was not adequately represented by each score in isolation. Each scale ranged from 0 to 5, with 0 being the best score.

This study aimed to investigate the interobserver agreement and reliability of each scale. It was hypothesized that, after training, observers would have excellent interobserver agreement and reliability for each scale and that these scales can be shown to have value for use in shelter and rescue settings and potentially in other contexts or types of research.

## 2. Materials and Methods

### 2.1. Rating Scales

Four ordinal rating scales were developed to assess different behavioural indicators of well-being in shelter cats. All four scales ranged from 0 to 5, with lower scores assumed to represent better well-being.

To assess signs of stress expressed through body language, slight adaptations to the Fear-Free Pets’ [8] Fear, Anxiety, and Stress (FAS) score were made. The version of the scale produced by Fear-Free Pets provides descriptions of different body language associated with various levels of well-being, but some descriptions are associated with more than one score. While this is valuable for the purpose for which the scale was created, it lacks the precision required to use this scale to monitor subtle changes over time or in response to specific interventions. For this reason, the scale was adapted to produce more definition between the levels of the scale (Table 1) and expand slightly on the operational definitions provided (Table 2).

A well-socialized cat’s level of physical interaction with a human might reasonably be expected to be reflective of their psychological well-being, as a very stressed cat may be more likely to appraise efforts to interact with them as a threat, while a relaxed cat may be more enthusiastic about these opportunities. This is particularly true for a cat with a history of affectionate interactions with humans, who can reasonably be expected to begin to show affectionate behaviour towards a human when they are experiencing reasonable psychological well-being. The Response to Petting (RTP) score was designed to convey the enjoyment and enthusiasm or fear and avoidance a cat expresses when faced with an attempt at petting (Table 3). While the scale includes scores that represent fear and avoidance, it is important to note that staff and volunteers are instructed not to attempt to touch cats if they are giving clear signs that they do not want to be touched. In these cases, staff and volunteers are instructed to record #N/A in the RTP data, making scores of 4 or 5 rare.

The Participation in Play (PIP) score was created to include an indicator of positive psychological well-being to supplement other measures that assess a negative psychological state. Play is a behaviour that is typically only exhibited when an animal’s other needs are met and is therefore a candidate for a positive psychological well-being indicator [9]. Enthusiasm of interaction with a toy as well as movement from their starting position were taken into consideration in development of the levels (Table 3).

It has been well-established that food intake is related to stress in cats [10,11]. This is not captured in the CSS. While most shelters already monitor the food intake of each individual cat, it is sometimes very complicated to interpret as it measures intake of both dry and wet food (sometimes of multiple types) over multiple meals daily. An ordinal variable that can summarise this information for ease of interpretation would be helpful. The Food Intake Summary (FIS) score was designed to represent the fraction of the food provided (both wet and dry) consumed over the previous 24 h (Table 3).

### 2.2. Training

A training system was designed to support the staff and volunteers using the ratings systems. A 1 h training was provided live through video for each scale (4 h total) and consisted of written and verbal descriptions, as well as examples. The trainings for the FAS, RTP, and PIP included two video examples of cats exhibiting behaviours characteristic of each level of the scale, and the training for the FIS included two photographic examples for each level of the scale from the food intake information collected on the shelter’s daily evaluation forms. All training sessions included time for questions from the participants.

### 2.3. Testing

Before being certified to contribute scores to the records of the cats in the shelter, it was required that participants complete a test after each training. The tests for the FAS, RTP, and PIP each consisted of 30 videos clips of up to 30 s and required the participants to rate the behaviour observed on the designated scale. The test for the FIS consisted of 30 photographs of the food intake information collected on the shelter’s daily evaluation forms and required the participants to provide a rating. Some cats appeared in more than one video or photograph. All scores were compared against those provided by the creator of the scale, which was treated as the reference standard.

### 2.4. Statistical Analysis

To investigate the interobserver agreement (the same value is achieved if a measurement is performed by different observers) [12] and reliability (the ability to differentiate among subjects) [12] of the scales, the test results of the first three cohorts trained were analyzed. Cohort 1 consisted of 6 participants, Cohort 2 consisted of 5 participants, and Cohort 3 consisted of 5 participants. Interobserver agreement of participants compared with the reference standard was assessed using a linear weighted κ, while interobserver reliability of participants compared with the reference standard was assessed using an Interclass Correlation Coefficient (ICC). Both methods were chosen based on the recommendations of de Vet [12], and the same methods having been used by van Haaften et al. [7], allowing for comparison. The linear weighted κ values were interpreted according to Landis and Koch [13], where <0.00 indicates poor agreement, 0.00–0.20 indicates slight agreement, 0.21–0.40 indicates fair agreement, 0.41–0.60 indicates moderate agreement, 0.61–0.80 indicates substantial agreement, and 0.81–1.00 indicates almost perfect agreement. ICC values were interpreted according to Koo and Li [14], where 0.00–0.49 indicates poor reliability, 0.50–0.74 indicates moderate reliability, 0.75–0.89 indicates good reliability, and 0.90–1.00 indicates excellent reliability. Each participant’s test scores for each rating scale were assessed against the reference standard, and averages were then taken across participants for both methods.

### 2.5. Case Studies

Several case studies have been presented to illustrate the practical applications of these systems.

## 3. Results

### 3.1. Interobserver Agreement and Reliability

Average linear weighted κ calculations revealed almost perfect agreement and average ICC revealed excellent reliability in all four rating scales (Table 4), although the minimum values indicated that not all participants achieved the same level of understanding.

### 3.2. Case Studies

Figure 1a–c illustrates different ways in which these ordinal rating systems can be of value to the shelter personnel responsible for safeguarding the psychological well-being of the cats in their care.

Figure 1a represents a cat who has a typical response to the shelter environment: initial stress response to shelter housing, with gradual improvement over an approximately 5-day habituation period [6]. The fact that all four scales reduce in a similar timeline in this case study supports their construct validity [15].

Figure 1b represents a cat who experienced a reduction in stress in response to an intervention. In this case, the cat expressed an initial stress response to shelter housing similar to what can be seen in Figure 1a, but in this case no significant reduction was observed during a comparable time frame. Instead, a reduction was observed only after the administration of gabapentin (days 5–7, illustrated on the graph with a green background). As these data represent only one case, it is not possible to definitively claim that the reduction in the four scales resulted from the gabapentin, but the case does show how these four scales can be used as outcome measures in scientific evaluation of the success of different interventions. This can help with case management decisions.

Finally, Figure 1c represents a cat who experienced an initial stress response to shelter housing without the degree of improvement shown in Figure 1a,b. The cat in Figure 1c exhibited a reduction in FIS, but increased food intake alone is not enough to interpret the psychological well-being of the cat to be acceptable. Interventions such as environmental management (e.g., out-of-cage space, social companion), formalized behaviour modification (utilizing desensitization and counter conditioning), and psychopharmaceuticals were employed (not represented on the graph), but none were successful in improving the cat’s psychological well-being. These data (in combination with information from the cat’s intake form and medical examination) were useful in determining that the cat was not appropriate for placement in a traditional home environment, and it was eventually placed through a barn cat program.

## 4. Discussion

This study presents four rating systems: the modified FAS, the RTP, the PIP, and the FIS. Results showed that all had almost perfect interobserver agreement and excellent interobserver reliability after a 1 h training session per rating system (4 h total).

This high degree of interobserver reliability and agreement ensures that these four tools can be used by multiple staff and volunteers to monitor the well-being of cats in shelters, as well as any progress in response to behavioural modification plans, while confidently overcoming the potential issues of non-standardised language through a common understanding of each level of the scales.

Patronek and Bradley [16] made a convincing case against the use of behaviour evaluations in shelters, and these four tools can arguably be considered behaviour evaluations. However, Patronek and Bradley argued against using the results of behaviour evaluations conducted in shelters to make predictions about how an animal would behave in a subsequent adoptive home, while the four tools proposed in this paper were created to monitor well-being and assess the success of strategies aimed at improving the lives of the cats while in the shelter. Therefore, these tools, if used for their intended purpose, do not suffer from the same risk of resulting in erroneous conclusions about how the cat might behave in another environment.

Beyond the shelter environment, the animal behaviour consulting community (working with owned animals in private homes) has been increasingly emphasizing the importance of collecting data to assess behavioural changes in animals being treated for various reasons. First, data reduce the phenomenon of imagining progress where there is none because the client is desperate to see signs of hope or please the consultant [17]. Next, data help keep clients from becoming frustrated by a perceived lack of progress when improvements are small and may potentially be missed if not tracking changes across time [18]. Finally, data allow consultants to compare the success of various treatment approaches across similar cases in order to select the best possible approaches for future cases [19]. Given sufficient training for pet owners, these four scales can also prove useful in the private behaviour consulting industry.

Ordinal rating scales are becoming more popular in broader animal behavior fields as a way to collect easily interpretable data. For example, Roney and Clark [20] used ordinal rating scales to investigate different aspects of working dogs’ performance such as motivation and stamina, while Shapiro et al. [21] used an ordinal rating system to quantify aggressiveness in bats. Similar rating systems are used in veterinary medicine to simplify complex results into more easily digestible information. For example, the International Renal Interest Society’s staging of chronic kidney disease [22] primarily uses fasted creatinine and/or SDMA concentrations to classify a dog or cat as meeting criteria for one of the stages on an ordinal scale. This simplified representation of the values allows for easier decision-making for treatment recommendations.

Within the field of cat behaviour, there are several ordinal rating systems that existed prior to this study. The most popular, as previously mentioned, is the CSS [6]. While this rating system is widely used, it has been criticized for not being sensitive enough to recognise certain expressions of stress (e.g., feigned sleep; [23]), lacking a published training guide [24], and variable or rarely reported interobserver reliability. Zeiler et al. [25] introduced a Demeanour Scoring system (referred to elsewhere as a Handling Scale score; [26]) to monitor behaviour of cats over time during hospitalization. This scale benefits from having reported interobserver agreement and reliability, with moderate interobserver agreement (κ = 0.606) and good interobserver reliability (ICC = 0.843). However, it was not truly an ordinal rating system, as the categorization within the scale was based on the response to eight structured behavioural tests with operationally defined levels. This means that the scale takes longer to rate and would likely require more training, making it less appropriate for a shelter environment. Additionally, the interobserver agreement and reliability were lower than the current study. It is unclear, however, how much training the three observers in Zeiler et al.’s study received, and as has been illustrated by Kessler and Turner [6], interobserver agreement and reliability can be closely linked to provision of training. Finally, two studies used an ordinal scale to describe a cat’s level of socialization. Vojtkovskáa et al.’s [27] 0–5 scale represented the behaviour exhibited by cats in response to approach and attempts at contact by a person, and used this score to investigate how these scores changed over time and their relationship with parameters such as length of stay. Jacobson et al.’s [28] scale classified cats into four groups based on their level of fear in general and in response to human contact both in shelter and in adoptive homes to investigate the post-adoption success of cats from a hoarding environment. Both authors’ socialization scales benefited from operational definitions, but neither were validated by comparing their scores to known socialization status prior to admission (as was performed for the Feline Spectrum Assessment [29,30,31]). Furthermore Zeiler et al.’s [25] scale did not investigate interobserver reliability (although they used only one rater, so this would have been impossible/unnecessary) and although Jacobson et al.’s [28] scale achieved substantial to almost perfect agreement (κ = 0.78–0.83) depending on the time point, scores were made retrospectively through interpretation of volunteer and adopter written descriptions and it is unclear if agreement would be as good based on live observation of behaviour.

The four scales presented in this manuscript offer many advantages for monitoring the behaviour of cats in shelters for the purposes of making inferences about their welfare and/or progress in response to interventions. First, each showed almost perfect interobserver agreement and excellent interobserver reliability. This is likely owing to the use of a formalized training program. The goal is to make this publicly available soon. Second, as has been shown through the case studies presented in this manuscript, these scales are sensitive enough to show change over time within individuals. Third, all four scoring systems are on the same scale (0–5, with 0 being the best and 5 being the worst), which makes interpretation and visual/graphical presentation easier and more intuitive. Finally, all four rating scales have analogues for use with dogs (unpublished) and similar scales for small domestic pets commonly held in shelters are in development. The advantage of having similar scales that can be used with different species is that this creates unity in methodology and understanding across the shelter, and between shelters, hopefully increasing the likelihood of use and elevating the perception of psychological well-being to parallel the importance of physiological well-being.

There are some limitations associated with the four scales presented in this study. First, the levels of each scale were not created through consultation with other industry professionals (internal or external). Consultation with and consensus among experts would help substantiate the face validity [15] of the scales being truly ordinal. However, the scales were created as a practical solution to the problem that appeared in the author’s shelter, namely that the language used to describe a cat’s behaviour was not standardised enough to truly gauge their welfare and/or progress in response to interventions, and forethought was not provided to the idea that this system would ultimately be deemed helpful enough to share with other organizations. Additionally, although the case studies presented show that the scales are sensitive enough to reflect change in a cat’s behaviour, they have not been used with a large enough sample size to compare the behaviour change in groups of cats with and without interventions. This type of analysis would help to substantiate the construct validity of each scale [15]. However, these scales lend themselves well to this type of investigation, and this is a recommended area of research for future studies; first confirming construct validity by investigating the impact of an intervention universally acknowledged to improve well-being, then using these scales to investigate the impact of rarely examined or novel interventions.

## 5. Conclusions

The scales presented in this study, the adapted FAS, RTP, PIP, and FIS, have standardised training systems, have shown good interobserver agreement and reliability, and are sensitive and valid enough to reflect positive changes in response to time and interventions in case studies. They can be useful for monitoring psychological well-being in animal shelters or academic studies, particularly for monitoring the impact of specific interventions.

## Figures and Tables

**Figure 1 animals-12-02897-f001:**
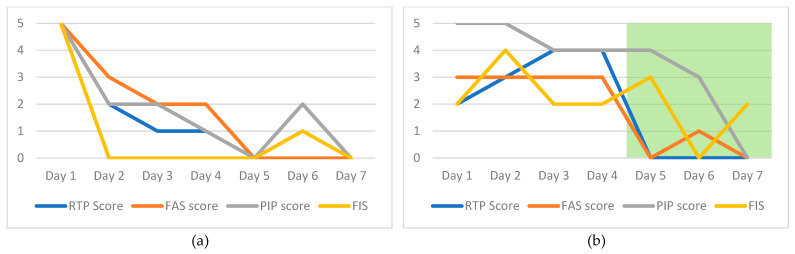
Evidence of an initial stress response to shelter housing, with gradual improvement through habituation (**a**); evidence of an initial stress response to shelter housing that reduced in response to administration of gabapentin (green block); (**b**); evidence of an initial stress response to shelter housing without improvement (with the exception of increased food intake) despite interventions (**c**). RTP: Response to Petting; FAS: Fear, Anxiety, and Stress; PIP: Participation in Play; FIS: Food Intake Summary.

**Table 1 animals-12-02897-t001:** Differentiation between adapted FAS categories.

FAS	High Stress Behaviours Present	Low Stress Behaviours Present	Classic Presentation
0	No signs	Many positive signs	Eagerly approaches tail up, relaxed posture, ears forward, eats treats/food in front of you
1	Some mild signs of stress	Mostly positive signs	Approaches tail up, relaxed posture/pupils/eyes, eats treats/food in front of you, ears may rotate a little, may hesitate from touch a moment, but then pushes into it
2	Consistent mild signs, some pronounced signs	Positive signs prevalent	Ears may rotate or be slightly to the side, body tense, avoids eye contact sometimes, vigilant, startles easily but recovers quickly, hesitance but then pushes into touch, licking lips
3	Pronounced signs prevalent, but severity low	Positive signs rare	Body tense (but not frozen), tail tucked, head up, some pupil dilation, startles easily, ears may be forward facing or rotating (but not flattened), licking lips, vigilant/watches hands, may sniff finger/eat treats, may hiss/growl, may breathe rapidly
4	Pronounced signs prevalent and severe	Positive signs likely absent	Ears flattened or to the side, pupils fully dilated, body/head pressed low, may hiss/growl, may breathe rapidly, may be frozen or fleeing
5	Pronounced signs prevalent and severe, includes aggression	Positive signs likely absent	Offensive/defensive aggression (lunging/swatting), may have same signs as score 4, or may be lacking other obvious signs of FAS

**Table 2 animals-12-02897-t002:** Definitions of high and low stress behaviours to aid in assigning an FAS score between adapted FAS categories.

Category	High Stress Behaviours	Low Stress Behaviours
Tail	Thrashing, tucked, twitching, down, wrapped	Up, relaxed on ground
Ears	Pressed down, airplane, sideways, rotating	Forward facing
Eyes	Wide, pressed shut, staring, averting gaze	Almond shaped, looking directly but not intensely
Pupils	Dilated, very small	Normal
Vocalizations	Hiss, growl, yowl, spit	Meow, chirp, mew
Body	Frozen, tense, standing with hind quarters lower than front, legs tucked under	Relaxed

Tips: Cats with an FAS of 3 or higher are considered to have questionable welfare. If you are deciding between a 2 and a 3, ask if the behaviour you are observing would make you concerned for their psychological well-being (if they were to exhibit it 24/7); never discount their initial behaviours, but rate based mostly on those behaviours they exhibit after habituating to your presence a bit; if a cat is friendly for most of an interaction, but extremely reactive for one procedure, score that cat a 2.

**Table 3 animals-12-02897-t003:** Instructions and operational definitions of the levels of the RTP, PIP, and FIS.

	RTP Description	PIP Description	FIS Description
**Instructions**	Open/enter the cat’s environment. Spend a few minutes getting them used to your presence and associating it with something positive (e.g., treats or play, whatever that cat responds to best). Then, attempt to pet the cat. How do they respond?	Begin your visit with a considerate approach. After giving the cat some time to become used to your presence (and some food/treats, if appropriate), attempt to engage him/her in play. Start with a low-key play strategy and adjust to more energetic play styles as the cat’s level of comfort suggests appropriate.	Use the food intake data collected over the last two meals (AM and PM) recorded on the form located on the cat’s clipboard to generate a score that matches up with the table below.
**Score 0**	Approach you and actively interact with you (e.g., tail up, rubbing against you, head-butting, cheerful vocalizations, may be purring)	Enthusiastically playing, moving around environment	Ate at least ¾ of either wet or dry at **both** meals
**Score 1**	Stay where they are but show clear signs of enjoying interaction (e.g., rolling around, clearly pushing their head into your hand, may be purring)	Tentatively playing, moving around environment	Ate at least ½ of either wet or dry at **both** meals
**Score 2**	Stay where they are but show mild signs of enjoyment (e.g., lift their chin slightly when you pet them, tense body)	Enthusiastically plays from a stationary position	Ate at least ¾ of either wet or dry at **either** meal
**Score 3**	Stay where they are but only tolerate petting (e.g., they do not move when you touch them, tense body)	Tentatively plays from a stationary position	Ate at least ½ of either wet or dry at **either** meal, or ¼ wet or dry at **both** meals
**Score 4**	Stay where they are but show signs of not wanting to be touched (e.g., flinching from your hand, flattening their ears, dilated pupils, tense body)	Watches toy but does not engage	Ate at least ¼ of either wet or dry at **either** meal, or ate well overnight but none/nibbling in day
**Score 5**	Flee from your attempts to touch, lunge at your attempts to touch	No interest in playing or retreats from attempts at play	Ate none/nibbling

**Table 4 animals-12-02897-t004:** Interobserver agreement and reliability.

	Linear Weighted κ			ICC		
	Average	Min, Max	Standard Deviation	Average	Min, Max	Standard Deviation
FAS	0.83	0.67, 0.97	0.077	0.94	0.85, 0.99	0.037
RTP	0.84	0.71, 0.93	0.060	0.94	0.88, 0.98	0.029
PIP	0.89	0.83, 0.97	0.043	0.95	0.85, 0.99	0.031
FIS	0.83	0.67, 0.93	0.091	0.90	0.73, 0.97	0.071

## Data Availability

The data presented in this study are available on request from the corresponding author. Since the intention is to make the training systems publicly available for use by other shelters or researchers, the data are not publicly available in an attempt to protect the quality of interobserver agreement and reliability of future trainings and assessments.

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
