# Peer review of "Beyond “Doing Better”: Ordinal Rating Scales to Monitor Behavioural Indicators of Well-Being in Cats"

_animals, 2022, doi:10.3390/ani12212897_

Round 1

Reviewer 1 Report

This is a description of a means of determining welfare of  caged cats. From the abstract and simple summary I expected  4 different scales, but really  it is one  5 point scale to rate each of 4 aspects of  welfare. Please rewrite. 

Methods

Table 2 hind quarters lower than front  This would describe a cat sitting What posture do you mean

L 123 1 hr ... for each  Scale  for  4 hr total or 1 hr total answered on l 196, but should be here

Define  interobserver agreement and reliability and construct validity 

 Not sure what was measured.  Are the results those of the people who took the training and then scored videos? If not please explain  how many cats they evaluated.

 What is a working cat environment ( mousing?)

 Fig 1 define initials in legend  RTP PIP  etc 

minor grammar point 2nd sentence  are not is

337 extra space in  c aregivers 

Author Response

Comments and Suggestions for Authors

Note: all line numbers refer to the originally submitted manuscript

This is a description of a means of determining welfare of  caged cats. From the abstract and simple  summary I expected  4 different scales, but really  it is one  5 point scale to rate each of 4 aspects of  welfare. Please rewrite. 

  • This change has been made

Table 2 hind quarters lower than front  This would describe a cat sitting What posture do you mean

  • Good point! I have clarified it to say “standing with hind quarters lower than front.” This posture is further clarified in the training program.

L 123 1 hr ... for each  Scale  for  4 hr total or 1 hr total answered on l 196, but should be here

  • This point has been further clarified at both line 123 and 196

Define  interobserver agreement and reliability and construct validity 

  • Definitions for interobserver agreement and reliability have been provided
  • A citation was provided for construct validity to conform with how the concept face validity had been referenced later in the manuscript

Not sure what was measured.  Are the results those of the people who took the training and then scored videos? If not please explain  how many cats they evaluated.

  • The section on testing has been given it’s own heading, and the sentence “Some cats were represented in more than one video or photograph” was added after the sentence “The tests for the FAS, RTP, and PIP each consisted of 30 videos clips of up to 30 seconds and required the participants to rate the behaviour observed on the designated scale. The test for the FIS consisted of 30 photographs of the food intake information collected on the shelter’s daily evaluation forms and required the participants to provide a rating.” It is my hope that this has better highlighted what data was collected and analysed.

What is a working cat environment ( mousing?)

  • I have clarified that it was a barn cat placement.

Fig 1 define initials in legend  RTP PIP  etc 

  • This has been added

minor grammar point 2nd sentence  are not is

  • I have changed the “is” to “are” on line 12

337 extra space in  c aregivers 

  • I believe this may be an artifact of the pdf – I have just examined the word document and the “c” and “a” in caregiver were right next to each other.

Reviewer 2 Report

50  'Noldus Observer' should have either TM  or ® after it. 

68  What was the advantage of the FAS over the CSS to assess the cat's body language? Or move lines 90-91 up to here.

83  with lower scores assumed to represent

102  ADD: well-being, or original socialization status toward humans, as a

Author Response

Comments and Suggestions for Authors

Note: all line numbers refer to the originally submitted manuscript

50  'Noldus Observer' should have either TM  or ® after it. 

  • This has been added

68  What was the advantage of the FAS over the CSS to assess the cat's body language? Or move lines 90-91 up to here.

  • I have moved lines 90-91 to where you suggested

83  with lower scores assumed to represent

  • This has been added

102  ADD: well-being, or original socialization status toward humans, as a

  • Opted to change the sentence in the following way instead. I believe it satisfies your concern while preserving the flow of the sentence.
    • A well-socialized cat’s level of physical interaction with a human might reasonably be expected to reflective of their psychological well-being, as a